# In vivo recellularization of xenogeneic vascular grafts decellularized with high hydrostatic pressure method in a porcine carotid arterial interpose model

Shunji Kurokawa[1], Yoshihide Hashimoto[2], Seiichi Funamoto[2,3], Kozue Murata[1,4], Akitatsu Yamashita[2], Kazuhiro Yamazaki[1], Tadashi Ikeda[1], Kenji Minatoya[1], Akio Kishida[2], Hidetoshi Masumoto[1]*

1 Department of Cardiovascular Surgery, Graduate School of Medicine, Kyoto University, Kyoto, Japan, 2 Department of Material-Based Medical Engineering, Institute of Biomaterials and Bioengineering, Tokyo Medical and Dental University, Tokyo, Japan, 3 Department of Biomedical Engineering, Osaka Institute of Technology, Osaka, Japan, 4 Institute for Advancement of Clinical and Translational Science, Kyoto University Hospital, Kyoto, Japan

* masumoto@kuhp.kyoto-u.ac.jp

**Data Availability Statement:** All relevant data are within the manuscript and its Supporting Information files.

## Abstract

Autologous vascular grafts are widely used in revascularization surgeries for small caliber targets. However, the availability of autologous conduits might be limited due to prior surgeries or the quality of vessels. Xenogeneic decellularized vascular grafts from animals can potentially be a substitute of autologous vascular grafts. Decellularization with high hydrostatic pressure (HHP) is reported to highly preserve extracellular matrix (ECM), creating feasible conditions for recellularization and vascular remodeling after implantation. In the present study, we conducted xenogeneic implantation of HHP-decellularized bovine vascular grafts from dorsalis pedis arteries to porcine carotid arteries and posteriorly evaluated graft patency, ECM preservation and recellularization. Avoiding damage of the luminal surface of the grafts from drying significantly during the surgical procedure increased the graft patency at 4 weeks after implantation (P = 0.0079). After the technical improvement, all grafts (N = 5) were patent with mild stenosis due to intimal hyperplasia at 4 weeks after implantation. Neither aneurysmal change nor massive thrombosis was observed, even without administration of anticoagulants nor anti-platelet agents. Elastica van Gieson and Sirius-red stainings revealed fair preservation of ECM proteins including elastin and collagen after implantation. The luminal surface of the grafts were thoroughly covered with von Willebrand factor-positive endothelium. Scanning electron microscopy of the luminal surface of implanted grafts exhibited a cobblestone-like endothelial cell layer which is similar to native vascular endothelium. Recellularization of the tunica media with alpha-smooth muscle actin-positive smooth muscle cells was partly observed. Thus, we confirmed that HHP-decellularized grafts are feasible for xenogeneic implantation accompanied by recellularization by recipient cells.

**Funding:** This work was supported by research grants from the Ministry of Education, Culture, Sports, Science and Technology, Japan (to T.I.) (17H04291) and Supporting Program for Interaction-based Initiative Team Studies (SPIRITS), Kyoto University (to K.M.). The funders had no role in study design, data collection and analysis, decision to publish, or preparation of the manuscript.

**Competing interests:** The authors have declared that no competing interests exist.

## Introduction

Cardiovascular disease is the first cause of death worldwide. According to the World Health Organization in 2016, 17.9 million people died from cardiovascular diseases, representing approximately 30% of total mortality [1]. Coronary artery disease (CAD) and peripheral artery disease (PAD) are major components of the increased mortality secondary to cardiovascular disease [2].

Revascularization surgery such as bypass grafting is a therapeutic approach for patients with CAD and PAD. However, considering the relatively small size of the target arteries, synthetic vascular prostheses made from expanded polytetrafluoroethylene, Dacron and other materials are not suitable for the revascularization of small caliber arteries (< 4 mm) due to the poor patency rate [3–5]. Autologous grafts such as internal mammary arteries, radial arteries or saphenous veins are commonly used for bypass grafting in small caliber arteries. Furthermore, the availability of autologous arterial grafts is rather limited. Although saphenous vein grafts are readily available, in 20–45% of patients requiring bypass surgeries, saphenous veins cannot be used due to prior vein surgeries, coronary or lower extremity revascularization or venous insufficiency [6, 7]. Therefore, alternative vascular grafts as a substitute for autologous grafts are anticipated.

Decellularized vascular grafts from xenogeneic origin are expected to be a novel resource for revascularization surgeries without the immunologic problems mainly caused by the xenogeneic cellular components [8]. We have been investigating a high hydrostatic pressure (HHP) method for decellularization [9] which can sufficiently deplete the donor tissue from cellular components while preserving extracellular matrices (ECM) proteins. The tissue tensile strength of decellularized grafts was superior to those obtained using chemical detergents [10]. Therefore, xenogeneic decellularized vascular grafts with a comparable caliber to human autologous vascular grafts can be prepared from animals and might address the unavailability of autologous grafts.

In the present study, we investigated the feasibility of HHP-decellularized vessels as small-caliber vascular grafts using a xenogeneic porcine carotid arterial interpose model through histological, ultrastructural, morphological and hemodynamic evaluations of the implanted vascular grafts.

## Materials and methods

All experimental procedures were approved by the Kyoto University Animal Experimentation Committee (#17542) and performed in accordance with the guidelines for Animal Experiments of Kyoto University, following the Japanese law and *the Guide for the Care and Use of Laboratory Animals* prepared by the Institute for Laboratory Animal Research, U.S.A. (revised 2011).

### Preparation of decellularized bovine dorsalis pedis arteries

Edible bovine legs were obtained from a local slaughterhouse (Tokyo Shibaura Organ, Tokyo, Japan). The dorsalis pedis arteries were harvested from the legs and washed in saline. The arteries were then packed in a plastic bag filled with saline and hydrostatically pressurized at 1000 MPa at 30˚C for 10 min using a cold isostatic pressurization machine (Dr. CHEF, Kobelco, Japan). The arteries were washed by continuous gradual shaking in saline supplemented with 0.2 mg/mL of Dnase I and 50 mM of magnesium chloride for 7 days and then in 80% ethanol for 3 days. Finally, after being washed, the arteries were stored in citric acid buffer (pH 7.4) at 4˚C until use.

## DNA quantification

The samples were freeze-dried and weighed (N = 5). Twenty mg of the samples were minced and digested overnight with 20 μg/mL proteinase K in 50mM of Tris-HCl, 1% sodium dodecyl sulfate, 100 mM of NaCl, and 25 mM of ethylenediaminetetraacetic acid -2Na solution at 55°C. DNA was isolated with a phenol/chloroform/isoamyl alcohol (25:24:1) extraction followed by ethanol precipitation. The amount of residual DNA was quantified using PicoGreen assay.

## Tensile test

The tensile test was conducted as previously described [10–12] using a universal testing machine (Autograph AG-X, Shimadzu) at crosshead speed of 0.1 mm/min. The samples were cut into dumbbell shape with a length of 35 mm and a width of 2 mm. The sample thickness was measured using micrometer with 2 μm accuracy before tensile test. Each specimen was preloaded to 0.01 N. Four specimens from each group were separately tested. The elastic modulus was estimated from the slope of liner fit to the stress-strain curve. The tests were performed following the longitudinal direction of the vessels, as testing in circumferential direction could not be done due to the small size of the vessels.

## Animal model and surgical procedure of implantation

Nine to ten-month-old female CLAWN miniature swine (25–27 kg; N = 9) were anesthetized with a ketamine (16 mg/kg body weight) and xylazine (1.6 mg/kg body weight) cocktail by intramuscular administration. Animals were intubated (Shiley Hi-Lo oral tracheal tube 7.5mm I.D. COVIDIEN JAPAN, Tokyo, Japan) and maintained on 1–2% isoflurane and 5 L oxygen using closed-circuit inhalation. During surgery, anesthesia depth and hemodynamic state were monitored by invasive blood pressure and electrocardiogram.

In each animal, a decellularized graft with HHP was implanted into the right carotid artery. After a midline neck incision, exposure of the right common carotid artery was performed for approximately 8 cm from carotid bifurcation. Heparin sodium (100 IU / kg) was administered before artery clamp. The right common carotid artery was interposed with the HHP decellularized graft for approximately 4 cm in an end-to-end fashion using 8–0 polypropylene continuous suture. In phase 1 group (N = 4), anastomoses were accomplished with similar surgical conditions to those used in human bypass grafting surgery (conventional condition). In phase 2 group (N = 5), anastomoses were carefully performed keeping the decellularized graft, especially its luminal surface, always wet with water (moist condition) and not touching its luminal surface. After anastomosis, blood flow was resumed. Blood flow velocity (time averaged maximum flow velocity) was measured by color Doppler ultrasound at the center of the implanted graft.

The wound was closed layer by layer. All animals were administered prophylactic cefazolin sodium intravenously. After implantation, no anticoagulants nor anti-platelet agents were administrated. Four weeks after implantation, decellularized grafts were explanted under general anesthesia and the animals were euthanized by intravenous bolus injection of potassium chloride (1–2 mEq/kg).

## Selective angiogram

An angiogram of the bilateral carotid arteries was performed by inserting 6 Fr guiding catheter (INTRODUCER II, TERUMO, Tokyo, Japan) through the femoral artery and a selective angiogram with a 4 Fr straight catheter (GLIDECATH, TERUMO) from the proximal end of

common carotid arteries. Iopamidol (OYPLOMIN®300, FujiPharma, Toyama, Japan; 100ml) was used as contrast agent. The angiograms were performed immediately after implantation surgery, 2 and 4 weeks after the surgery, respectively.

### Intravascular ultrasound (IVUS)

IVUS was performed parallel to each angiogram, respectively. VISIONS PV .018 (Philips Japan, Tokyo, Japan) was used for IVUS transducer and VOLCANO (Philips Japan) was used for the data acquisition. IVUS was performed for bilateral carotid arteries (contralateral side of the implanted side was used as a control). The data was recorded from the distal to proximal anastomosis site of the implanted graft to evaluate intraluminal stenosis and morphology. Data was analyzed using Image J software (version 1.50i, National Institutes of Health, Bethesda, MD) [13].

### Histological analysis

Explanted grafts were incised in longitudinal direction and fixed in 4% paraformaldehyde for 48 hours. After fixation, explanted grafts were divided into proximal and distal parts and subsequentially embedded in paraffin. In each part, sections with 6 μm thickness were prepared consecutively and subjected to hematoxylin and eosin, Elastica van Gieson (EVG), Sirius red and von Kossa staining respectively. Immunostaining of von Willebrand factor for endothelium (anti-von Willebrand Factor antibody, Abcam, Cambridge, UK, 1:3000), α-smooth muscle actin for smooth muscle cells (anti-alpha smooth muscle actin antibody, Abcam, 1:500), CD45 for leukocytes (rabbit polyclonal anti-CD45 antibody, ab10558, Abcam, 1:10000), CD68 for porcine macrophages (mouse anti pig macrophages antibody, clone BA4D5, Bio-Rad Antibodies, Puchheim, Germany, 1:50) and CD90 (human/porcine/canine CD90/Thy1 antibody, R&D Systems, Minneapolis, MN, 1:20) were performed, respectively. Sirius red staining sections were observed using polarized light microscope (BX51; Olympus, Tokyo, Japan). For the quantification of collagen and elastin after decellularization, Sirius red and EVG staining were used respectively (2 samples for each staining), and quantified by automatic measurement of an all-in-one microscope (BZ-X800, Keyence, Osaka, Japan).

### Scanning electron microscopy (SEM)

Native bovine dorsalis pedis arteries decellularized grafts before implantation, native porcine carotid arteries and explanted grafts were fixed into a 2.5% Glutaraldehyde-2% Paraformaldehyde-0.1 M Phosphate Buffer solution (pH 7.4) for 24 hours at 4˚C, respectively. After fixation, samples were immersed into 1% osmium tetroxide for 2 hours at 4˚C and were dehydrated by graded ethanol (50%, 60%, 70%, 80%, 90%, 95% and 100%) for 30 min. Subsequently, the samples were dried and coated with a thin layer of platinum palladium using an ion sputtering device (Eiko Corp.,Tokyo, Japan). The sample was examined with a Hitachi S-4700 scanning electron microscope (Hitachi, Tokyo, Japan).

### Statistical analysis

All data analyses were performed using JMP version 11.2.0 (SAS Institute, Cary, NC, USA). Statistical analysis of the data was performed with unpaired t-tests or Fisher's exact test for 2 groups. $P < 0.05$ was considered significant. Values are reported as means ± SD.

## Results

### Physical and biochemical characteristics of decellularized vascular grafts with HHP method

The macroscopic view of the decellularized bovine dorsalis pedis artery with HHP method is shown in Fig 1A. The amount of residual DNA was measured to quantify the efficiency of cell removal. The amount of residual DNA of decellularized grafts was 15.85 ± 2.15 ng/mg which was significantly lower compared with that of bovine arteries, 4451.0 ± 351.7 ng/mg (P < 0.0001) (Fig 1B).

Tensile test was performed to measure physical strength of decellularized grafts. Decellularized grafts exhibited 0.25 ± 0.05 MPa in early phase modulus of elasticity (mainly indicating

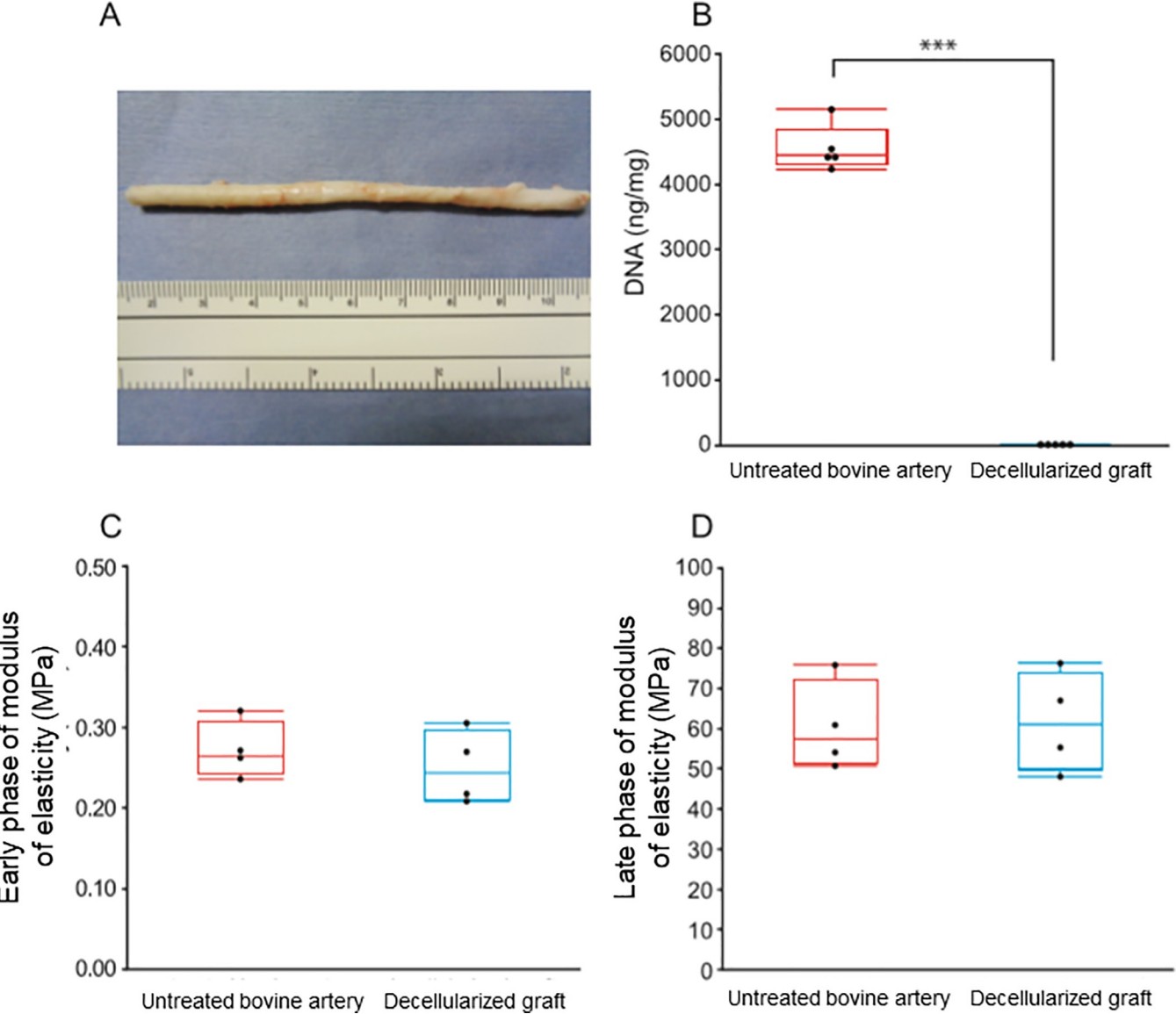

**Fig 1. Characteristics of decellularized grafts with HHP method.** (**A**) A representative macroscopic view of a decellularized bovine vessel with HHP method. HHP, high hydrostatic pressure. Upper scale of measure indicates centimeters, and lower indicates inchs, respectively. (**B**) Quantification of residual DNA. ***P<0.001. (**C**) Early phase of modulus of elasticity. (**D**) Late phase of modulus of elasticity.

stress sustained by elastin fibers) [10, 14] and 61.69 ± 12.49 MPa in late phase modulus of elasticity (mainly indicating stress sustained by collagen fibers) [10, 14], while untreated bovine artery showed 0.27 ± 0.04 MPa and 60.42 ± 11.14 MPa (Fig 1C, 1D), respectively. There were no significant differences in values of both phases (P = 0.50 in early phase modulus of elasticity, and P = 0.88 in late phase modulus of elasticity). We also analyzed the relationship between strain and stress and found that the patterns of curves are similar between untreated and decellularized arteries (S1 Fig). The contents of collagen and elastin after HHP-decellularization were histologically quantified in average as 51.5% and 19.0% of the whole graft area, respectively (S2 Fig).

## Implantation surgery of HHP-decellularized vascular grafts

We started our experiments under surgically equivalent conditions to those of usual human revascularization surgeries in preparation and anastomoses of autologous vascular grafts such as saphenous vein grafts (Phase 1; N = 4). We experienced that all 4 implanted grafts were occluded at 4 weeks after implantation (0/4; 0% patency in Phase 1). We considered that the major reason of the occlusion might be damage of the intima and exposure of basement membrane attributed by the drying of a lumen side which may affect antithrombogenicity and patency [15], therefore we then modified the condition of the anastomoses keeping them moist, avoiding the lumen of grafts to be dried as much as possible (Phase 2; N = 5) (Fig 2A and 2B; S1 Video). In phase 2, we confirmed that all 5 grafts were patent at 4 weeks after implantation (5/5; 100% patency in Phase 2) (Phase 1 vs Phase 2; P = 0.0079). There was no significant difference in stime averaged maximum flow velocity just after anastomosis between Phase 1 and Phase 2 (Fig 2C and 2D).

## Evaluations for implanted graft patency and morphology

In all cases of Phase 2, moderate stenoses of grafts were observed in the proximity of both anastomosis sites by selective angiogram (Fig 3A). Intimal thickening of the corresponding regions was also confirmed by IVUS (Fig 3B). Stenosis ratios of proximal and distal anastomotic regions were 49.4 ± 0.12% and 51.4 ± 0.04%, respectively. We optically evaluated all explanted grafts at 4 weeks after implantation. Luminal side of all explanted grafts were macroscopically smooth, but some explanted grafts were accompanied by a small amount of thrombi (Fig 3C). No aneurysmal change was observed in phase 2 cases.

## Histological evaluations for implanted grafts after implantation

Hematoxylin and eosin staining revealed that HHP-decellularized graft did not contain cell nuclei indicating sufficient decellularization by HHP. On the other hand, whole layer recellularization of the grafts was confirmed at 4 weeks after implantation (Fig 4A). Elastica van Gieson and Sirius red staining exhibited that elastin layer, tunica media and stratified elastin layers were fairly preserved in HHP-decellularized grafts and grafts at 4 weeks after implantation; and comparable to those of native arteries before decellularization at tunica media (Fig 4B). Polarized microscopical observations for striated elastin layer revealed that collagen I deposition was preserved among the elastin layers before implantation and newly produced collagen III were deposited at the same region after implantation (S3 Fig).

Immunostaining for von Willebrand Factor-positive endothelium revealed that the intima of implanted grafts was fairly covered by an endothelial cell layer throughout the graft (Fig 4C). α-smooth muscle actin (αSMA)-positive vascular smooth muscle cells were observed amongst the tunica media (Fig 4D). These results indicate that the HHP-decellularized vascular grafts were recellularized by host-derived vascular cells in accordance with the anatomical allocations of native arteries.

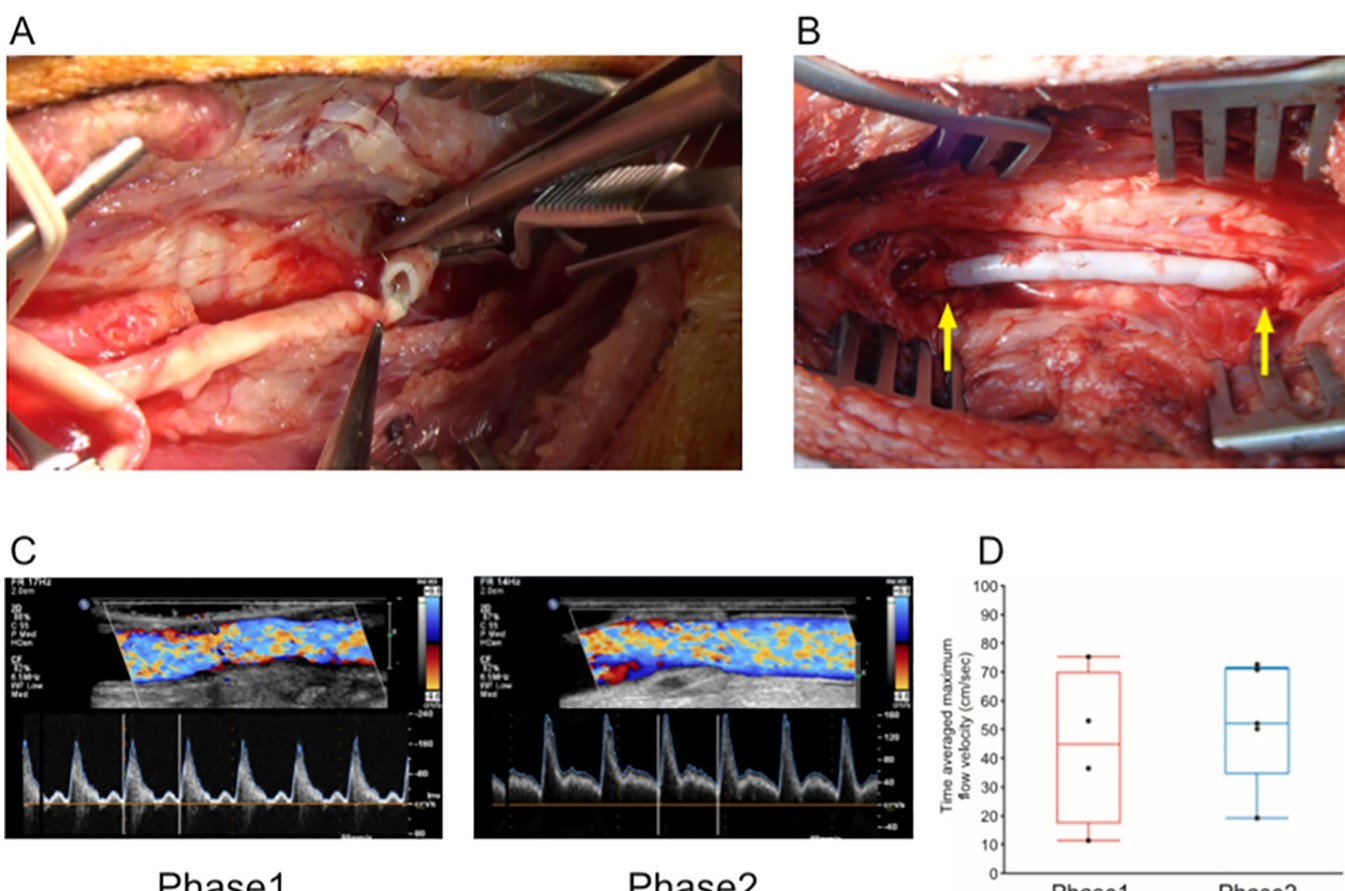

**Fig 2. Implantation of decellularized graft into porcine carotid artery.** (**A**) (**B**) A representative surgical view during (A) and after (B) anastomosis in moist condition (Phase 2). Yellow arrows indicate anastomosis sites. (**C**) Representative images of color Doppler ultrasound at the center of implanted graft (left; Phase 1, right; Phase 2). (**D**) Time averaged maximum flow velocity.

We evaluated the stenotic regions in the proximities of the proximal and distal anastomoses. Hematoxylin and eosin staining exhibited that the hypertrophic regions were filled with proliferated cellular components. Immunostaining for von Willebrand Factor revealed thin endothelial cell layer covering the luminal surface. Immunostaining for αSMA showed that the stenotic region mainly consisted of proliferated smooth muscle cells located between surface endothelial cell layer and internal elastic lamina (Fig 4E).

Von Kossa staining showed a small deposition of calcified nodules close to the suture line which were not observed in any other region of the graft (S4 Fig). We evaluated the infiltration of inflammatory cells toward the implanted grafts. CD45, CD68 and CD90 immunostaining revealed that the implanted grafts were not infiltrated with inflammatory cells such as macrophages or fibroblasts except for some regions close to the luminal surface (S5 Fig). However, we also found that inflammatory cells and fibroblasts were considerably accumulated at the adjacent region of the implanted grafts.

## SEM

The ultrastructure of luminal surface of vessels was evaluated by SEM. The luminal surface of native bovine artery (graft animal) was covered with endothelium exhibiting a cobblestone-like appearance (Fig 5A). After decellularization by HHP processes, the luminal surface of

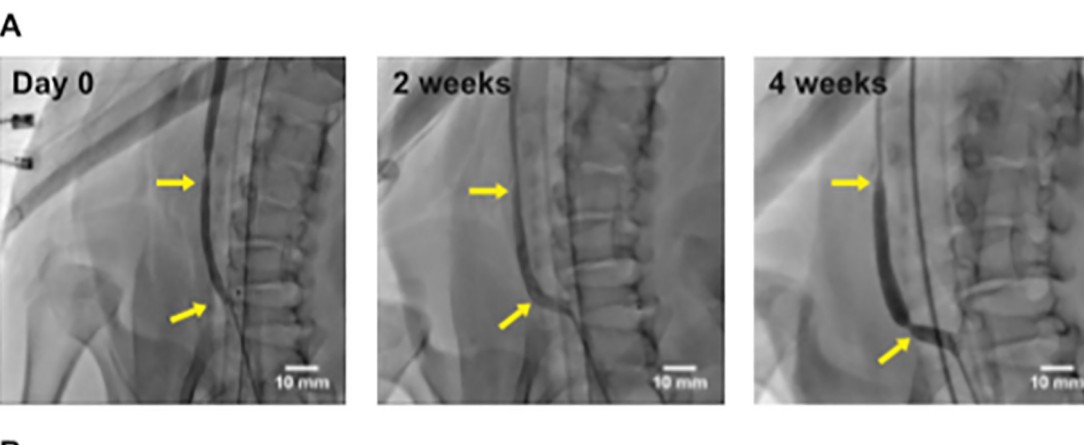

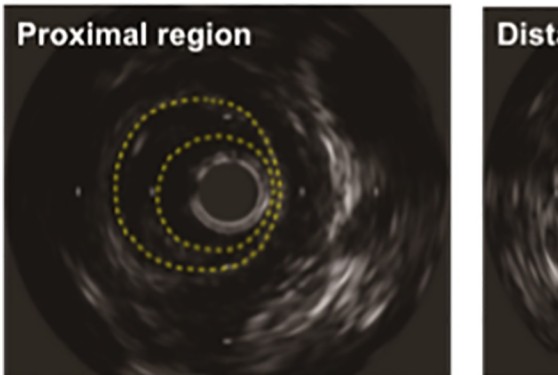
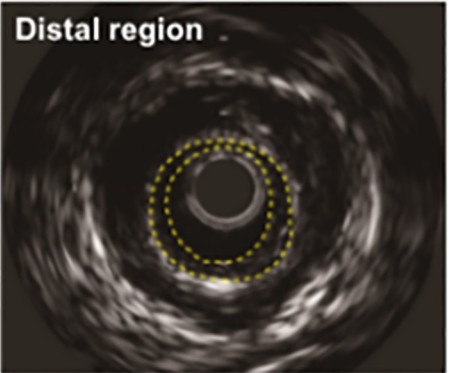

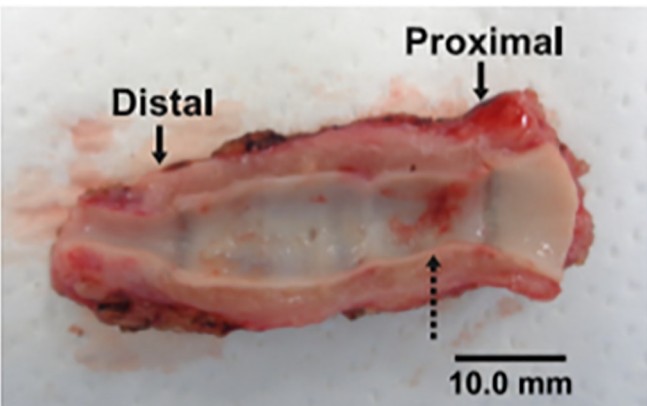

**Fig 3. Evaluation of the implanted graft patency and morphology.** (**A**) Representative images of angiography of implanted grafts at just after implantation (left), and 2 weeks (middle) and 4 weeks (right) after implantation, respectively. Yellow arrows indicate anastomosis sites. (**B**) Representative images of IVUS. Left; proximal region, right; distal region. (**C**) A representative macroscopic view of explanted graft. Arrows indicate anastomosis sites. Dotted arrow indicates thrombi.

decellularized grafts exhibited acellular smooth surface without endothelium (Fig 5B). The luminal surface of decellularized bovine graft implanted at porcine carotid artery for 4 weeks showed endothelium with cobblestone-like appearance by fair recellularization similar with those in bovine and porcine native arteries (recipient animal) (Fig 5A, 5C and 5D).

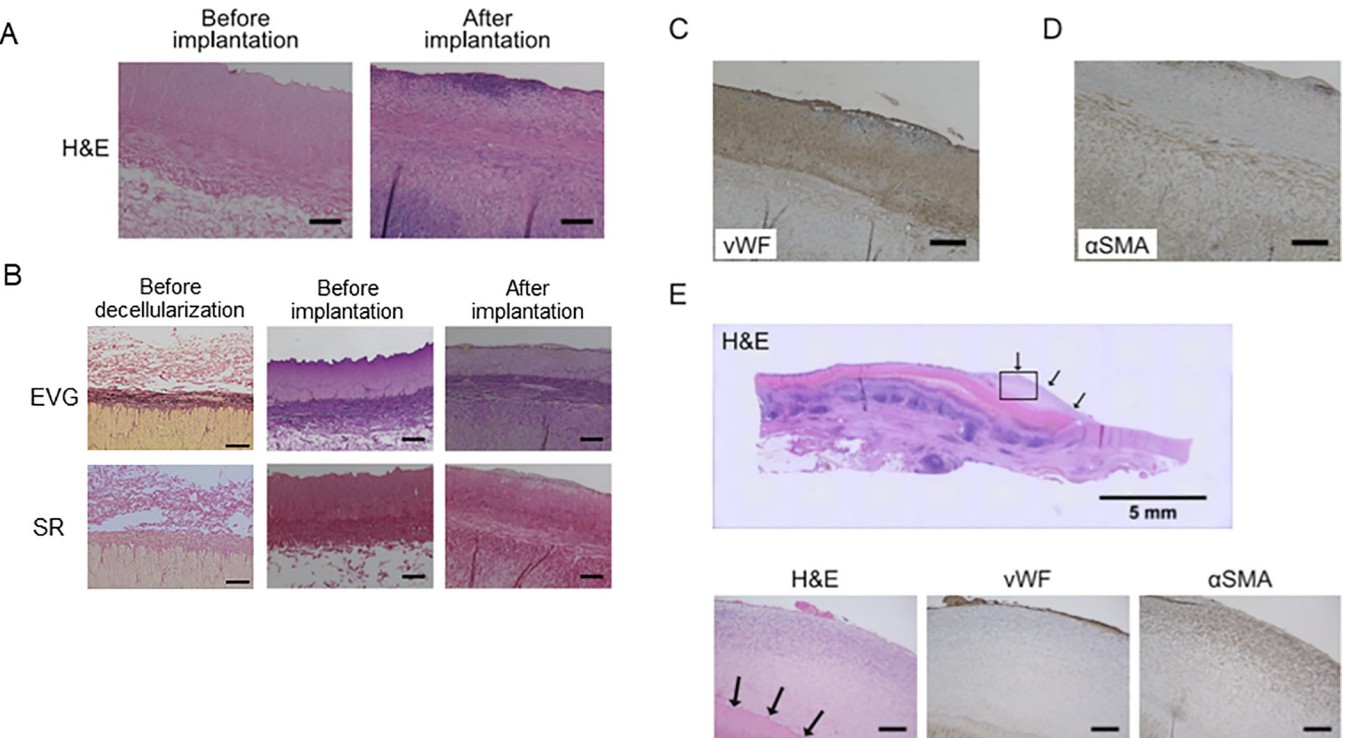

**Fig 4. Histological evaluations for decellularized grafts after implantation. (A)** Representative Hematoxylin and Eosin (H&E) staining before implantation (left) and 4 weeks after implantation (right). Scale bars = 200 μm. (**B**) Representative Elastica van Gieson (EVG) staining and Sirius red (SR) staining, before implantation (left) and 4 weeks after implantation (right). Scale bars = 200 μm. (**C**) Representative von Willebrand Factor (vWF) immunostaining at 4 weeks after implantation. Scale bar = 200 μm. (**D**) Representative α-smooth muscle actin staining at 4 weeks after implantation. Scale bar = 200 μm. € H&E staining for representative stenotic regions close to proximal anastomoses at 4 weeks after implantation. Black arrows indicate hypertrophic region (top). H&E staining (bottom left), vWF immunostaining (bottom middle), and α-smooth muscle actin (αSMA) immunostaining (bottom right) for hypertrophic region indicated with square in top figure. Scale bars = 200 μm.

## Discussion

In the present study, we confirmed that HHP-decellularized small-caliber vascular grafts can be recellularized by xenogeneic implantation with acceptable patency. The intimal layer of implanted grafts was covered by host-derived endothelial cells which could maintain the antithrombogenicity of the graft. These results, acquired from a large animal model with similar size of target vessels in human revascularization surgeries, might indicate the feasibility using HHP-decellularized vascular grafts in xenogeneic implantation surgeries.

For patients requiring revascularization surgery, unavailability of autologous bypass grafts may lead to loss of an opportunity for appropriate therapy and consequent poor prognosis. Although cryopreserved arterial allografts can be a surrogate for autologous small-caliber grafts with acceptable long-term patency, the availability is rather limited because of the shortage of donors and insufficient operation of organ / tissue banks to preserve and provide allografts [16–18]. The present study was designed to address this healthcare problem through validating the feasibility of xenogeneic implantation of HHP-decellularized vascular grafts using bovine arterial grafts and a porcine carotid arterial interpose model.

Numerous methods for decellularization of living tissues have been reported so far [19–21]. In previous reports of the implantation of decellularized small-caliber vascular grafts by chemical or biological decellularization methods, insufficient outcomes such as graft occlusion, thrombus formation, and intimal proliferation throughout the graft were observed [22, 23].

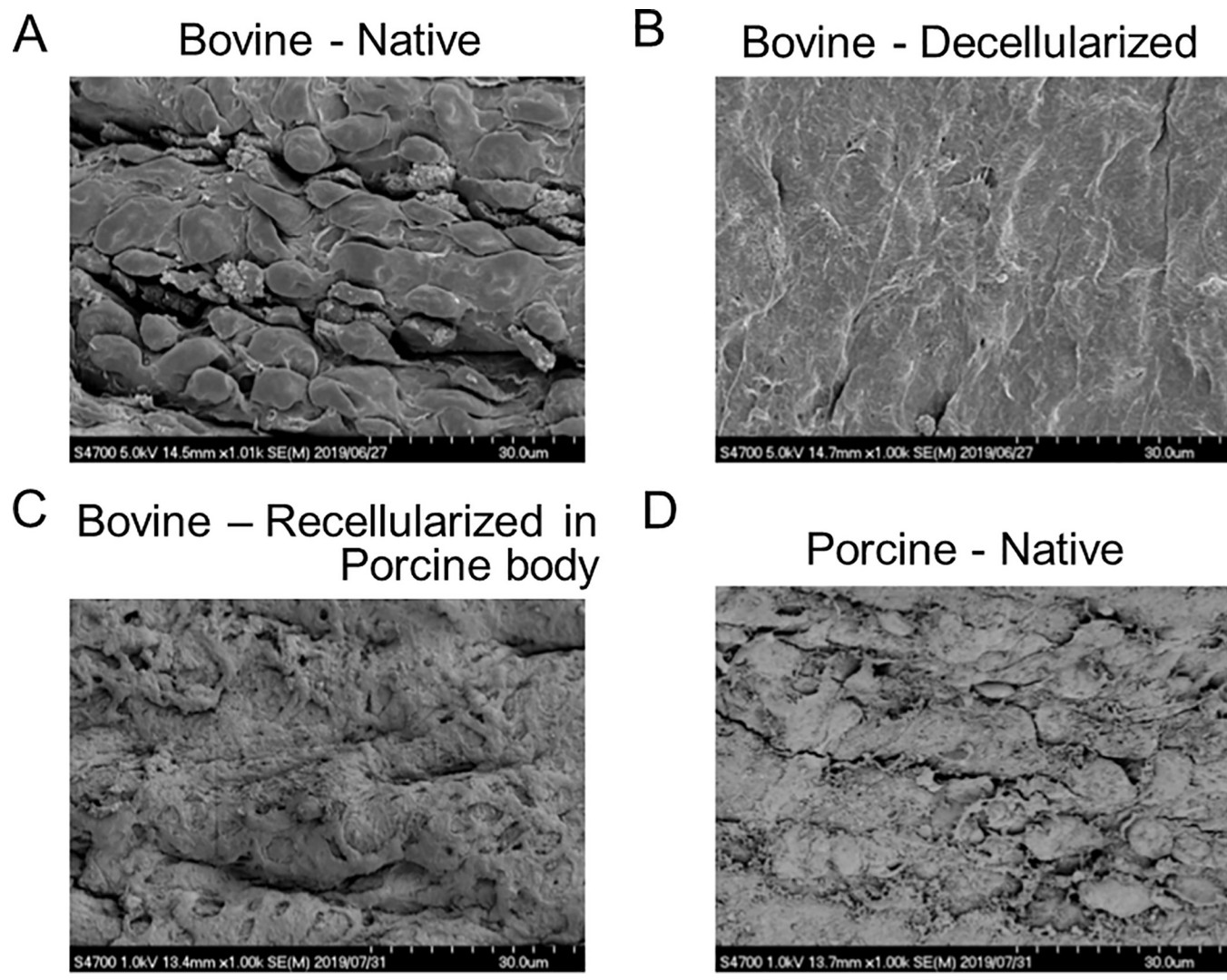

**Fig 5. Evaluation of endothelial formation by scanning electron microscopy.** (**A**) Representative luminal surface of untreated bovine artery. (**B**) Representative luminal surface of decellularized bovine artery. (**C**) Representative luminal surface of explanted graft. (**D**) Representative luminal surface of native porcine carotid artery.

We previously reported that the HHP decellularization, a novel method utilizing a physical basis can almost completely wash out the cellular components of porcine aorta and radial arteries with preserved mechanical properties such as elastic modulus [9]. In the present study, we decellularized bovine dorsalis pedis arteries by HHP and confirmed that the efficiency of cellular wash-out was >99% calculated by residual DNA amounts compared to that of untreated bovine dorsalis pedis artery. Residual DNA amounts in the present study (15.85 ± 2.15 ng/mg) satisfied the recommended criteria of successful decellularization as <50 ng/mg [24]. In mechanical tensile tests, there was no significant difference between HHP-decellularized and native arteries in the early and late elastic modulus representing the elastin phase and the collagen phase even though some of the protein components would be removed during the decellularization process [25]. This result indicates that the physical strength of HHP-decellularized bovine arteries was not impaired by the HHP. The late phase modulus of elasticity in the present study was much higher than the previous results of HHP-decellularized

aortic intima-media (slice of aorta) [10] suggesting that HHP-decellularized vessels in the present study can be used as a cardiovascular biomaterial under high blood pressure and we could experimentally confirm the feasibility in the present study of implantation to porcine carotid arteries. Histological evaluations for HHP-decellularized arteries revealed that HHP did not damage the ECM structures of native arteries. Taken together, HHP method was proven to maintain structural and mechanical function of the arteries with enough depletion of cellular components.

In our previous research, we compared mechanical properties of decellularized aortic intima-media with HHP and sodium dodecyl sulphate (SDS)-based chemical method and found that mechanical stress of HHP-treated samples was much higher than that of SDS-treated samples [10]. On the other hand, we confirmed in the study that there are many voids in SDS-decellularized aortic intima-media which might allow early *in vivo* recellularization after implantation. It might be possible that SDS-decellularized tissues may be used for applications in environments with low blood pressure such as vein substitutes with efficient recellularization. In other words, there would be a possibility that lower void formation of HHP-decellularized grafts would not be suitable for early *in vivo* recellularization. It would be desirable to select appropriate decellularization methods in accordance with the purpose and the site of the implantation. We should also recognize that the extent of infiltration of inflammatory cells including macrophages and fibroblasts are not less than those in previously reported implantation experiments of chemically decellularized valves or aortic tissues to porcine body [26, 27] even high efficiency of decellularization by HHP. The accumulation of macrophages at the adjacent region of implanted grafts would mediate both inflammatory tissue damage and reparative process such as remodeling of extracellular matrices scaffolds [27] in accordance with the polarization status of accumulated macrophages [28] which should be further investigated in our future studies.

The cellular source of *in vivo* recellularization in the present study would be recipient cells (not residual host cells) considering extremely high decellularization efficiency of the HHP method. Histological evaluations and SEM for decellularized grafts at 4 weeks after xenogeneic implantation revealed that the luminal surface was covered by the recipient's endothelial cells, and smooth muscle cells were infiltrated into the media layer. These results indicate that the recellularization took place in an organized manner according to the original structure of the arteries. On the other hand, we simultaneously observed irregular structural reconstruction such as calcification around the suture line and intimal hyperplasia which mainly consisted of smooth muscle cells proliferating at proximities of proximal and distal anastomoses. The progression of intimal hyperplasia might be a cause of the occlusion of the grafts. Even though the hyperplasia and calcification did not affect the blood flow of the implanted graft at the observation period of the present study, histological changes and patency should be followed up for longer period in our future study. Considering that the stenotic sites existed in close proximity of suture lines, there would be a possibility that turbulence of the blood flow at suture lines, possibly due to the calcification or the existence of suture threads, affected the stenosis formation. The establishment of preventive strategies for the stenosis is crucial towards clinical implementation of the HHP-decellularized grafts as an allogeneic graft source in the future.

In the present study, implantation surgeries were performed in 2 phases which were different in surgical conditions especially in moisture conditions of the grafts (Phase 1; conventional condition, Phase 2; no luminal surface touch of the graft keeping a moist condition). Flow patterns in both phases immediately after implantation did not differ to each other, indicating that the qualities of anastomoses were not different in the two phases. However, the patency was significantly lower in Phase 1 compared to that in Phase 2 at 4 weeks after implantation.

In Phase 1, all implanted grafts resulted in thromboembolism, whereas all grafts in Phase 2 were patent with a small amount of thrombi without use of any postoperative antiplatelet or anticoagulants drugs. These results suggest that the luminal surface of HHP-decellularized vascular grafts possesses fair antithrombogenicity when the intimal surface was not damaged by intraoperative grasping or drying. Although a careful manipulation would be required in bypass grafting surgeries, HHP-decellularized vascular grafts might hold promise as a novel vascular graft without antiplatelet agents or anticoagulants in the future.

There are several limitations in the present study. 1) The presented histological results are rather qualitative because of difficulties in quantification using commercially available porcine-specific antibodies. Establishment of quantitative evaluations for recellularization and immune response in our future work might further validate the presented results. 2) The observation period should be longer as some of the previous experiments of decellularized small diameter vascular grafts [29], to further validate the long-term feasibility and patency of HHP-decellularized small diameter grafts.

## Conclusion

Xenogeneic HHP decellularized graft showed feasible capacity for recellularization and vascular remodeling without thrombogenicity. HHP decellularized vascular grafts may be utilized as new medical products for revascularization surgeries.

## Supporting information

**S1 Fig. Curves between stress and strain of arteries.** Curves between stress and strain in untreated bovine arteries (**A**) and decellularized arteries (**B**) at early phase (upper) and late phase (lower). Results of 3 independent experiments are shown.
(TIF)

**S2 Fig. Representative images of Sirius red (SR) staining and Elastica van Gieson (EVG) staining of decellularized grafts.** Scale bars = 500 μm.
(TIF)

**S3 Fig. A representative polarized microscopy for decellularized grafts.** Before (left) and after (right) implantation. Scale bars = 200 μm.
(TIF)

**S4 Fig. A representative von Kossa staining for grafts after implantation.** Scale bars = 5 mm (top), 100 μm (bottom).
(TIF)

**S5 Fig. A representative immunostaining for inflammatory cells and fibroblasts.** CD45 (**A**), CD68 (**B**) and CD90 (**C**) immunostaining for grafts after implantation. Left: lower magnification, right: higher magnification. Scale bars = 200 μm.
(TIF)

**S1 Video. Surgical procedure of xenogeneic implantation of decellularized grafts in moist condition (phase 2).**
(MP4)

**S1 File. Minimal data set.**
(XLSX)

## Acknowledgments

We thank Mr. S. Miyake, Ms. Y. Matsubara, Mr. H. Koda and Ms. K. Furuta (Kyoto University) for technical assistance. We thank Dr. Laura Yuriko González Teshima (Kyoto University) for reviewing English sentences.

## Author Contributions

**Conceptualization:** Shunji Kurokawa, Tadashi Ikeda, Kenji Minatoya, Akio Kishida, Hidetoshi Masumoto.

**Data curation:** Shunji Kurokawa, Yoshihide Hashimoto, Seiichi Funamoto, Kozue Murata, Akitatsu Yamashita, Akio Kishida, Hidetoshi Masumoto.

**Formal analysis:** Shunji Kurokawa, Yoshihide Hashimoto, Seiichi Funamoto, Kozue Murata, Tadashi Ikeda, Akio Kishida, Hidetoshi Masumoto.

**Funding acquisition:** Tadashi Ikeda, Kenji Minatoya.

**Investigation:** Shunji Kurokawa, Yoshihide Hashimoto, Seiichi Funamoto, Kozue Murata, Akitatsu Yamashita, Kazuhiro Yamazaki, Tadashi Ikeda, Kenji Minatoya, Akio Kishida, Hidetoshi Masumoto.

**Methodology:** Shunji Kurokawa, Yoshihide Hashimoto, Akio Kishida, Hidetoshi Masumoto.

**Project administration:** Shunji Kurokawa, Akitatsu Yamashita, Kazuhiro Yamazaki, Tadashi Ikeda, Kenji Minatoya, Akio Kishida.

**Supervision:** Seiichi Funamoto, Akitatsu Yamashita, Kazuhiro Yamazaki, Tadashi Ikeda, Kenji Minatoya, Akio Kishida, Hidetoshi Masumoto.

**Validation:** Shunji Kurokawa, Yoshihide Hashimoto, Seiichi Funamoto, Kozue Murata, Akitatsu Yamashita, Kazuhiro Yamazaki, Tadashi Ikeda, Kenji Minatoya, Akio Kishida, Hidetoshi Masumoto.

**Visualization:** Shunji Kurokawa, Kozue Murata, Hidetoshi Masumoto.

**Writing – original draft:** Shunji Kurokawa, Hidetoshi Masumoto.

**Writing – review & editing:** Shunji Kurokawa, Yoshihide Hashimoto, Tadashi Ikeda, Kenji Minatoya, Hidetoshi Masumoto.

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
