## [Decision Letter · Decision Letter 0]

14 Dec 2020

PONE-D-20-30796

In vivo recellularization of xenogeneic vascular grafts decellularized with high hydrostatic pressure method in a porcine carotid arterial interpose model

PLOS ONE

Dear Dr. Masumoto,

Thank you for submitting your manuscript to PLOS ONE. After careful consideration, we feel that it has merit but does not fully meet PLOS ONE’s publication criteria as it currently stands. Therefore, we invite you to submit a revised version of the manuscript that addresses the points raised during the review process. Please make sure that the quantification of the immune cells have been performed appropriately to evaluate graft response and that cells found in repopulated grafts have been stained with a specific antibodies to ensure that recellularization of the grafts can be attributed to invading cells from the recipient.

We look forward to receiving your revised manuscript.

Kind regards,

Seda Kizilel, PhD

Academic Editor

PLOS ONE

Journal Requirements:

1 Please ensure that your manuscript meets PLOS ONE's style requirements, including those for file naming. The PLOS ONE style templates can be found at

Reviewers' comments:

Reviewer's Responses to Questions

**Comments to the Author**

1. Is the manuscript technically sound, and do the data support the conclusions?

Reviewer #1: Yes

Reviewer #2: Yes

2. Has the statistical analysis been performed appropriately and rigorously? 

Reviewer #1: Yes

Reviewer #2: Yes

3. Have the authors made all data underlying the findings in their manuscript fully available?

Reviewer #1: Yes

Reviewer #2: Yes

4. Is the manuscript presented in an intelligible fashion and written in standard English?

Reviewer #1: No

Reviewer #2: Yes

5. Review Comments to the Author

Reviewer #1: The paper describes the preparation of decellularized xenogeneic vascular grafts by the method of high hydrostatic pressure. The grafts are tested for mechanical properties in vitro and then implanted in vivo to test patency at 2 and 4 weeks. The authors did modify the suturing and handling protocol to improve outcomes. Overall, the paper is interesting but it needs major revision. Detailed comments are outlined below.

1. The whole paper needs to be revised for English, especially the introduction as it is distracting to read. Specifically, there are incomplete sentences, improper sentence structure, grammatical errors, and wrong word choices.

2. In the introduction or discussion, there is no description of other studies that use HHP for similar purposes. How and why is this study different, what are some advantages and disadvantages of HHP and why is HHP especially beneficial for decellularizing vascular grafts.

3. Additionally in the discussion, it would be helpful if the authors give context of their results related to results from other groups using either HHP or different de-cellularization methods for the purpose of creating xenogeneic vascular grafts. Are the results from this work comparable, better, worse? There is a general lack for context for the presented results, so it is hard to judge their significance.

4. More discussion is also needed on patency as a function of time and how the authors’ results compare to other studies of decellularized small diameter vascular grafts.

5. Limitations of the technique and the achieved grafts should be discussed.

6. Could the authors back up with a reference their suggestion that stenosis could be due to suture lines? Could they offer more discussion?

7. It is not clear what is meant by early vs late phase modulus. Could the authors elaborate more?

8. The results are overall very qualitative in nature and some more quantification and image analysis will strengthen the paper.

Reviewer #2: The manuscript addresses the decellularization of small caliber bovine blood vessels using high-hydrostatic pressure. The authors present an optimization of the surgical procedure to avoid initial graft failure, most probably attributed to the drying of the luminal region of the decellularized vessels. Importantly, challenges related to this study are addressed, and possible solutions to avoid graft failure in the long term are presented in the Discussion. Despite this, the manuscript lacks quantitative data, and requires improvement before acceptance.

Comments:

- The authors consider that recellularization of the grafts can be ascribed to invading cells from the recipient, because residual DNA content found after the decellularization procedure is low. To make this statement, cells found in repopulated grafts must be stained with a specific species (porcine) antibody.

- Although staining of histological sections shows valuable data, the content of elastin and collagen in the grafts (pre-implant) must be evaluated quantitatively.

- Although fibroblasts did not invade the graft luminal area, it seems that a considerable recruitment of fibroblasts to its peripheral area occurred. Could the deposition of a fibrotic capsule around the graft impair its functionality overtime? Quantification of the presence of immune cells, namely macrophages (and their polarization state) would be beneficial to understand the extent of the response caused by the graft. Comparison with results reported in the literature for other decellularization methods would be desirable at this stage.

- Stress-strain curves presented in Supporting Information must be presented for a significant number of replicates, for example as means and standard deviation.

6. PLOS authors have the option to publish the peer review history of their article (what does this mean?). If published, this will include your full peer review and any attached files.

Reviewer #1: No

Reviewer #2: No

---

## [Author Response · Author response to Decision Letter 0]

30 May 2021

First of all, we deeply appreciate the important comments from reviewers which greatly improved the quality of our manuscript. Taking all of your feedback into consideration, we conducted additional experiments and revised our manuscript. 

For Reviewer #1:

1. The whole paper needs to be revised for English, especially the introduction as it is distracting to read. Specifically, there are incomplete sentences, improper sentence structure, grammatical errors, and wrong word choices. 

Thank you for your comment. We apologize for this point; a complete revision of the manuscript was done by an English language editor (especially Introduction section). 

2. In the introduction or discussion, there is no description of other studies that use HHP for similar purposes. How and why is this study different, what are some advantages and disadvantages of HHP and why is HHP especially beneficial for decellularizing vascular grafts. 

Thank you for the comment. We have reported several researches of implantation experiments of HHP-decellularized vascular grafts. In our first report, we performed allogeneic implantation of decellularized porcine aorta (Funamoto et al. Biomaterials. 2010;31:3590-5; Ref. #9). After that study, we focused on small diameter vessels which would be beneficial for revascularization surgeries for ischemic diseases (Negishi et al. J Artif Organs. 2011;14:223-31 / Negishi et al. J Tissue Eng Regen Med. 2015;9:E144-51; Ref. #11 / Negishi et al. J Biomed Mater Res A. 2017;105:1293-1298) and performed syngeneic or xenogeneic implantation into rat carotid arteries. In all researches, we could confirm sufficient patency and cell attachment on the luminal surface after implantation. However, all experiments of small diameter vessels were performed in rodents, and implantation experiment in a large animal (similar to human) with similar size of target vessels of human revascularization surgeries was anticipated. The present study is our first study to confirm patency and recellularization of HHP-decellularized small diameter vessels using a large animal model which might be beneficial for clinical implementation of the HHP-based decellularization. We added discussion regarding this point at DISCUSSION section (line 335 - 336, page 19).

In our previous research (Wu et al. Interact Cardiovasc Thorac Surg. 2015;21:189-94; Ref. #10), we compared mechanical properties of decellularized aortic intima-media (slice of aorta) with HHP and SDS-based chemical method and found that mechanical stress of HHP-treated samples was much higher than that of SDS-treated samples (Figure 1 for Reviewer #1). The tensile strength of HHP-treated samples was higher than those of untreated human heart valves. The late phase modulus of elasticity in the present study was much higher than the previous results of HHP-decellularized aortic intima-media (61.7 ± 12.5 vs 6.2 ± 0.8 vs MPa), which suggests that HHP-decellularized vessels in the present study can be used as a cardiovascular biomaterial under high blood pressure and we could experimentally confirm the feasibility in the present study of implantation to porcine carotid arteries. This would be the largest advantage of the HHP method. We added discussion regarding this point at DISCUSSION section (line 364 – 368, 373 - 376, page 21). 

On the other hand, we confirmed in the previous study (#10) that there are many voids in HE staining of SDS-decellularized aortic intima-media which might allow early in vivo recellularization after implantation. Therefore, SDS-decellularized aortic intima-media may be used for applications in environments with low blood pressure such as coverage materials or vein substitutes with efficient recellularization. These results might show the feasibility of various cardiovascular biomaterials with different properties by selecting the appropriate decellularization method. We added discussion regarding this point at DISCUSSION section (line 376, page 21 – line 383, page 22). 

3. Additionally in the discussion, it would be helpful if the authors give context of their results related to results from other groups using either HHP or different de-cellularization methods for the purpose of creating xenogeneic vascular grafts. Are the results from this work comparable, better, worse? There is a general lack for context for the presented results, so it is hard to judge their significance. 

Thank you for the comment. As answered in Comment #2, we compared mechanical properties of decellularized aortic intima-media (slice of aorta) with HHP and SDS-based chemical method in our previous research (Ref. #10) and found that mechanical stress of HHP-treated samples was much higher than that of SDS-treated samples. It would be feasible to mention that the mechanical properties of HHP-decellularized tissues would be much better compared to those decellularized with chemical methods. On the other hand, as also mentioned in Comment #2, it would be possible to say that the efficiency of early in vivo decellularization might be worse compared to those with chemical methods due to low voids after HHP-decellularization. 

 We apologize that the lack for this context in the initial submission and added discussion regarding this point at DISCUSSION section (line 373, page 21 – line 383, page 22). 

4. More discussion is also needed on patency as a function of time and how the authors’ results compare to other studies of decellularized small diameter vascular grafts. 

Thank you for your comment. In literature review, we found several researches of xenogeneic implantation of decellularized small diameter vascular grafts in large animal models and confirmed that most of the researches set 1 month or longer observation period. The longest observation period was found in implantation of SDS-decellularized porcine carotid arterial xenografts to carotid arteries of goats. The research observed the patency using color Doppler ultrasonography at predetermined intervals (1 week, 1 month, 3 months, 6 months, 12 months after implantation; two animals at each interval) and confirmed 95% of patency (Kim et al. Int J Artif Organs. 2007;30:44-52; new #29). We recognize that the observation period of the present study would be rather short and we should prepare experiments with longer observation period in our future study. 

 We added a limitation regarding this point (line 429 - 431, page 24). We added a reference (new #29). 

5. Limitations of the technique and the achieved grafts should be discussed. 

Thank you for the comment. As answered in Comment #2, we confirmed in the previous study (#10) that there are many voids in HE staining of SDS-decellularized aortic intima-media which might allow early in vivo recellularization after implantation. In another word, there would be a possibility that lower void formation of HHP-decellularized grafts would not be suitable for early in vivo recellularization and might be a limitation of the HHP technique. It would be desirable to select appropriate decellularization methods in accordance with the purpose and the site of the implantation. We added discussion regarding this point at DISCUSSION section (line 376, page 21 – line 383, page 22). 

6. Could the authors back up with a reference their suggestion that stenosis could be due to suture lines? Could they offer more discussion? 

Thank you for the important comment. As shown in a previous study (new #27), calcification at the suture line might be commonly observed after implantation of decellularized grafts (Figure 2 for Reviewer #1). There might be a possibility that the deformation of vascular luminal surface due to the calcification causes turbulence of blood flow and subsequent stenosis. 

 We added discussion regarding this point at DISCUSSION section (line 407, page 23). We added a new reference (new #27). 

7. It is not clear what is meant by early vs late phase modulus. Could the authors elaborate more? 

Thank you for the comment. As described in Ref. #10 and new #14, the stress is primarily sustained by the elastin fibers at the beginning of the strain; this is known as the early phase. Then the collagen fibers become the major stress-bearing components during the later phase (Figure #3 for Reviewer #1). 

According to the comment, we added detailed description about early and late modulus in RESULTS section (line 213 - 215, page 13). We added a new reference (new #14). 

8. The results are overall very qualitative in nature and some more quantification and image analysis will strengthen the paper. 

Thank you for the comment. We agree that the results especially histological experiments are qualitative because of the nature of the present study as the reviewer pointed out. We believe that the establishment of quantitative evaluations for histological evaluations in our future work might further validate presented results. We added a limitation regarding this point (line 425 - 429, page 24). 

For Reviewer #2:

1. The authors consider that recellularization of the grafts can be ascribed to invading cells from the recipient, because residual DNA content found after the decellularization procedure is low. To make this statement, cells found in repopulated grafts must be stained with a specific species (porcine) antibody. 

Thank you for the comment. According to the professional comment from the reviewer, we additionally performed immunostaining using porcine-specific swine leukocyte antigen (SLA) antibody (Mouse anti Pig SLA Class I antibody, clone JM1E3, Bio-Rad Laboratories, Inc.; 1:20 dilution). As shown in Fig. 1 for Reviewer #2, we could observe SLA-positive cells at the intimal surface of the vascular graft after implantation, whereas we could not observe SLA-positive cells at the surface of native bovine vessel. The result indicates that the cells found in recellularized tissue are originated from host pigs. 

2. Although staining of histological sections shows valuable data, the content of elastin and collagen in the grafts (pre-implant) must be evaluated quantitatively. 

Thank you for the comment. Following the reviewer’s comment, we conducted additional staining experiments for pre-implant grafts and quantified the contents of elastin and collagen by EVG staining and Sirius red staining, respectively and added the results in RESULTS section (line 219 - 221, page 14). We added descriptions related to the additional staining in MATERIALS AND METHODS section (line 181, page 11 – line 184, page 12). We added representative EVG staining and Sirius red staining images of decellularized vessels in new Supplemental Figure 2. 

3. Although fibroblasts did not invade the graft luminal area, it seems that a considerable recruitment of fibroblasts to its peripheral area occurred. Could the deposition of a fibrotic capsule around the graft impair its functionality overtime? Quantification of the presence of immune cells, namely macrophages (and their polarization state) would be beneficial to understand the extent of the response caused by the graft. Comparison with results reported in the literature for other decellularization methods would be desirable at this stage. 

Thank you for the comment. According to the comment, we conducted additional immunostaining for porcine-specific CD68 as a porcine macrophage marker. As shown in new Supplemental Fig. 5B, macrophages are accumulated at the adjacent native tissue and intimal surface of the implanted grafts. We clearly described that inflammatory cells including macrophages and fibroblasts are accumulated at the adjacent region of implanted grafts in RESLTS section (line 295, page 17 – line 299, page 18). 

By literature review, we found 2 papers of decellularized tissue implantation into pig heart or vessels and histological evaluations for post-implant macrophage infiltration (Fig. 2 for Reviewer #2). Numata et al. reported allogeneic implantation of cryopreserved pulmonary valve and found less macrophage infiltration in chemically decellularized pulmonary valves using Triton X compared to those in valves without decellularization (no quantitative evaluation) (J Heart Valve Dis. 2004;13:984-90; new #26). Paniagua Gutierrez et al. reported implantation of SDS-decellularized porcine aortic root to porcine abdominal aorta and found that MAC-positive macrophages are infiltrated below the neointima of the explanted tissue (no quantitative evaluation) (Tissue Eng Part A. 2015;21:332-42; new #27). In both studies, the extent of macrophage infiltration was comparable with our present results shown in Supplemental Fig. 5B. We added new references (#26, 27). 

We thank the reviewer to indicate this point which enabled us to add more information and discussion about the cellular type of accumulated inflammatory cells and inflammatory processes mediated by the graft. Now we recognize the possibility that the inflammatory processes might affect the formation of fibrotic capsule around the graft which might impair the vessel function. Another possible biological process mediated by macrophages would be the reparative remodeling of extracellular matrices scaffolds as discussed in Ref. #27. Although we could not evaluate the polarization state of accumulated macrophages this time because of the unavailability of suitable antibody for porcine-specific M1 or M2 macrophages, it might be possible that both inflammatory tissue damage mediated by M1 macrophages and reparative process mediated by M2 macrophages simultaneously take place after implantation. We added discussions regarding this point in DISCUSSION section (line 384 - 391, page 22). We added a reference regarding macrophage polarization (new #28). 

 On the other hand, it was technically difficult for us to precisely quantify the infiltration of macrophages because of the relatively low signal/noise ratio of immunostaining even after rigorous optimization of staining conditions possibly because of relatively low quality of commercially available porcine-specific antibody, and we added just qualitative figure instead of quantification of macrophage infiltration. We recognize the qualitative tendency of data presentation in the present study and added a limitation regarding this point (line 425 - 429, page 24). 

 We added methods of CD68 immunostaining in MATERIALS AND METHODS section (line 176 - 180, page 11) (We also added information about immunostaining for CD45 and CD90 which are not shown in the initial submission). We added representative results of CD68 immunostaining as new Supplemental Fig. 5B. 

4. Stress-strain curves presented in Supporting Information must be presented for a significant number of replicates, for example as means and standard deviation. 

Thank you for the important comment. According to the comment, we revised Supplemental Fig. 1. Because of difficulties in showing data as means and standard deviation because of different extent of tissue strain which is independently determined in each sample, we showed results of 3 independent experiments for untreated bovine arteries and decellularized arteries instead. 

We believe that we were able to address the reviewers' comments satisfactorily. We hope that our manuscript is now fully acceptable for publication in PLoS One. We truly appreciate your kind editing efforts on our manuscript.

Sincerely,

Hidetoshi Masumoto, MD, PhD

---

## [Decision Letter · Decision Letter 1]

22 Jun 2021

In vivo recellularization of xenogeneic vascular grafts decellularized with high hydrostatic pressure method in a porcine carotid arterial interpose model

PONE-D-20-30796R1

Dear Dr. Masumoto,

We’re pleased to inform you that your manuscript has been judged scientifically suitable for publication and will be formally accepted for publication once it meets all outstanding technical requirements.

We expect that you improve the presentation of additional figures with higher magnification in Supplemental Figure 5 to make cell infiltration results more clear to the readers.

Kind regards,

Seda Kizilel, PhD

Academic Editor

PLOS ONE

Additional Editor Comments (optional):

Reviewers' comments:

Reviewer's Responses to Questions

**Comments to the Author**

1. If the authors have adequately addressed your comments raised in a previous round of review and you feel that this manuscript is now acceptable for publication, you may indicate that here to bypass the “Comments to the Author” section, enter your conflict of interest statement in the “Confidential to Editor” section, and submit your "Accept" recommendation.

Reviewer #1: All comments have been addressed

Reviewer #2: All comments have been addressed

2. Is the manuscript technically sound, and do the data support the conclusions?

Reviewer #1: Yes

Reviewer #2: Yes

3. Has the statistical analysis been performed appropriately and rigorously? 

Reviewer #1: Yes

Reviewer #2: Yes

4. Have the authors made all data underlying the findings in their manuscript fully available?

Reviewer #1: Yes

Reviewer #2: Yes

5. Is the manuscript presented in an intelligible fashion and written in standard English?

Reviewer #1: Yes

Reviewer #2: Yes

6. Review Comments to the Author

Reviewer #1: (No Response)

Reviewer #2: The authors addressed most of my questions, and the addition of required experimental results followed by a further critical discussion of the significance of the work provided by the authors improved the quality of the manuscript. I believe that the presentation of additional figures with higher magnification in Supplemental Figure 5 would be important to make cell infiltration results more clear to the readers.

7. PLOS authors have the option to publish the peer review history of their article (what does this mean?). If published, this will include your full peer review and any attached files.

Reviewer #1: No

Reviewer #2: No

---

## [Editor Report · Acceptance letter]

14 Jul 2021

PONE-D-20-30796R1 

In vivo recellularization of xenogeneic vascular grafts decellularized with high hydrostatic pressure method in a porcine carotid arterial interpose model 

Dear Dr. Masumoto:

I'm pleased to inform you that your manuscript has been deemed suitable for publication in PLOS ONE. Congratulations! Your manuscript is now with our production department. 

Kind regards, 

on behalf of

Dr. Seda Kizilel 

Academic Editor

PLOS ONE